# Treatment outcomes of cutaneous leishmaniasis due to *Leishmania aethiopica*: A systematic review and meta-analysis

**Abebaw Yeshambel Alemu** [1,2,3] *, **Lemma Derseh**[2], **Mirgissa Kaba**[4], **Endalamaw Gadisa**[3], **Kassahun Alemu**[2]

**1** Department of Pediatrics and Neonatal Health Nursing, College of Health Sciences, Debre Tabor University, Debre Tabor, Ethiopia, **2** Department of Epidemiology and Biostatistics, Institute of Public Health, College of Medicine and Health Sciences, University of Gondar, Gondar, Ethiopia, **3** Armauer Hansen Research Institute, Addis Ababa, Ethiopia, **4** School of Public Health, Addis Ababa University, Addis Ababa, Ethiopia

* yeshambelabebaw@gmail.com

## Abstract

**Data Availability Statement:** All relevant data are within the paper and its Supporting information files.

### Background

*Leishmania aethiopica* is a unique species that causes cutaneous leishmaniasis (CL), and studies evaluating treatment outcomes for this condition reported inconsistent findings. This study aimed to summarize the evidence on treatment outcomes of CL caused by *L. aethiopica* to support decisions or propose further study.

### Methods

We searched PubMed, Scopus, and ScienceDirect. In addition, we searched grey literature on Google Scholar and performed manual searching on the reference list of articles. Two authors did the screening, selection, critical appraisal, and data extraction. With the narrative synthesis of evidence, we performed a random effects model meta-analysis using the *metaprop* package in Stata 17. We did sensitivity and subgroup analyses after assessing heterogeneity using the I-squared test and forest plots. The funnel plot and Egger's test were used to assess publication bias.

### Results

The review included 22 studies with 808 participants, and the meta-analysis included seven studies with 677 participants. Most studies documented treatment outcomes with antimonial monotherapy, and only one study reported outcomes with combination therapy. The overall pooled proportion of cure was 63% (95% CI: 38–86%). In the subgroup analysis, systemic antimonial monotherapy showed a cure rate of 61%, and the proportion of cure was 87% with topical therapy. Topical therapy showed a better cure for the localized clinical phenotype. A cohort study documented a cure rate of 94.8% with combination therapy for the localized, mucocutaneous, and diffuse clinical phenotypes. The pooled proportion of

**Funding:** The author(s) received no specific funding for this work. However, the Skin Health Africa Research Program project financially supported the first author in the form of a student stipend, but the project did not decide on the design, analysis and reporting of this study.

**Competing interests:** The authors have declared that no competing interests exist.

**Abbreviations:** CL, Cutaneous Leishmaniasis; DCL, Diffuse Cutaneous Leishmaniasis; IM, Intramuscular; IV, Intravenous; LCL, Localized Cutaneous Leishmaniasis; MA, Meglumine Antimonate; MCL, Mucocutaneous Leishmaniasis; NOS, Newcastle Ottawa Scale; QoD, Every Other Day; RoB, Risk of Bias Assessment; SSG$^V$, Sodium Stibogluconate.

unfavourable outcomes was partial response (19%), relapse (17%), discontinuation (19%), and unresponsiveness (6%).

## Conclusions

The pooled proportion of cure is low with antimonial monotherapy. Despite limited evidence, combination therapies are a promising treatment option for all clinical phenotypes of CL caused by *L. aethiopica*. Future high-quality randomized control trials are needed to identify effective monotherapies and evaluate the effectiveness of combination therapies.

## Introduction

Cutaneous leishmaniasis (CL) is a category one emerging and uncontrolled neglected tropical disease (NTD) caused by obligate intracellular protozoa; genus *Leishmania* [1]. More than 20 *Leishmania* species reported worldwide, but *L. tropica* and *L. major* cause most CL in Asia, Europe, and Africa [2]. *Leishmania aethiopica* is a unique species that causes almost all CL in Ethiopia and the Mount Elgon area of Kenya [3]. Two hyrax species (*Heterohyrax brucei* and *Procavia capensis*) and sandfly species (*Phlebotomus longipes* and *Ph. pedifer*) are the known reservoirs and vectors of *L. aethiopica*, respectively [4]. *L. aethiopica* causes three clinical phenotypes including localized cutaneous leishmaniasis (LCL), mucocutaneous leishmaniasis (MCL) and diffuse cutaneous leishmaniasis (DCL) [5]. While the LCL is the commonest type, MCL is less common, and DCL is rare [3].

The responses of CL to anti-leishmaniasis therapy vary by causative species [1, 5]. A meta-analysis of randomized control trial (RCT) studies showed that *L. major* respond better with photodynamic therapy. This study also reported that, for *L. tropica*, thermotherapy is more effective compared with intramuscular sodium stibogluconate (IM-SSG$^V$) [6]. Because of species-level unresponsiveness, SSG$^V$ is becoming ineffective across all *Leishmania* species [7]. Similarly, *L. aethiopica* has also shown variation for different anti-leishmaniasis treatment options. Three Kenyan with confirmed *L. aethiopica* get treated with a high dose (18 to 20 mg/kg twice daily) of intravenous-SSG$^V$ (IV-SSG$^v$) for 30 days had shown disappearance of parasite from skin slit smear (SSS) and from culture after 14 to 27 days, and clinical healing of lesion and no recurrence at 3 to 18 months follow-up [8]. Studies report variation in treatment outcomes of CL due to *L. aethiopica* among immigrants [9–11]. A Belgian traveler who returned from northern Ethiopia had a complete resolution of the lesion with IM meglumine antimonate [11]. In addition, an Eritrean immigrant in Germany showed healing with IV-amphotericin B [10]. As such, three Ethiopian immigrants in Israel had complete healing of lesion with a combination of 15% paromomycin sulfate and 12% methyl benzethonium chloride ointment [9].

In the national leishmaniasis treatment guideline of Ethiopia, withhold treatment, intralesional SSG$^v$, or cryotherapy are recommended for LCL treatment. In this guideline, systemic SSG$^v$ and paromomycin are recommended for MCL and DCL treatment, as topical agents are for unresponsive LCL [12]. A randomized phase II trial conducted among LCL cases in Ethiopia showed a 69% reduction in lesion size at 16 weeks with a four-week topical application of Shiunko ointment compared to only 22% in the placebo group [13]. Additionally, observational studies evaluating outcomes of CL due to *L. aethiopica* report cure rates ranging from 12.5% to 85% across different clinical phenotypes [14–17]. Studies done in Ethiopia also report unfavourable treatment outcomes, such as relapse, change in the treatment regimen, treatment extension, and failure [14–17]. Furthermore, two cohort studies done in Ethiopia report

predictors of treatment outcomes of CL due to *L. aethiopica* [17, 18]. Among LCL cases treated with six cycles of intralesional-SSG[v] (IL-SSG[v]), a higher chance of cure was positively associated with being male, increasing age and SSS grade +1 and +2 [18]. With 28 days of oral miltefosine treatment, MCL cases showed lower odds of relapse on day 180 [17]. LCL cases treated with six cycles of IL-SSG [18] and all clinical phenotypes treated with 28 days oral miltefosine showed an increased lesion size [17].

Yet, randomized control trial studies that assessed the effectiveness of different treatment options hardly exist elsewhere to base treatment decisions of CL due to *L. aethiopica* [19]. Some observational studies and very few non-randomized clinical trials that documented treatment outcomes for this species report inconsistent findings to base clinical decisions. Therefore, this systematic review and meta-analysis aimed to summarize evidence on the treatment outcomes of *L. aethiopica*-caused CL. This is important to support clinical decision-making and suggest potential alternative treatment options for patient care.

## Materials and methods

### Reporting

The protocol was registered in the PROSPERO database [**CRD42022306698**], and reported using the Preferred Reporting Items for Systematic Review and Meta-analysis (PRISMA) checklist 2020 [20] (S1 Checklist).

### Eligibility criteria

We included studies done on human beings that reported the responses of *L. aethiopica* to anti-leishmaniasis therapy, published in English but without restriction to the year of publication and study design. As imported CL cases due to *L. aethiopica* were reported in non-endemic countries, we did not limit studies by country. Systematic reviews, comments to the editor, conference proceedings, and abstracts without full-length articles were excluded.

### Search strategy

The Mnemonic PEO (Problem, Exposure, Outcome) was used to organize searching. A comprehensive search of PubMed, Scopus, and ScienceDirect databases was performed using Boolean operators 'AND' and 'OR'. In PubMed and Scopus, we use the search terms: "Leishmaniasis" OR "Oriental sore" OR "Cutaneous Leishmaniasis" OR "Diffuse Cutaneous Leishmaniasis" OR "Old World Cutaneous Leishmaniasis" OR "Mucocutaneous Leishmaniasis" OR "*L. aethiopica*" OR *Leishmania aethiopica*" AND "Cryotherapy" OR "Heat therapy" OR "Thermotherapy" OR "Systemic therapy" OR "Localized therapy" OR "Oral therapy" OR "Miltefosine" OR "Sodium stibogluconate" OR "Antimoniate" OR "Glucantime" OR "Meglumine antimoniate" OR "Liquid nitrogen therapy" OR "Paromomycin" OR "Amphotericin b" OR "Withholding treatment" OR "Conservative treatment" OR "Laser therapy" AND "Treatment outcome*" OR "Treatment failure" OR "Treatment Cure" OR "Relapse" OR "Unresponsiveness" OR "Non-responsiveness" OR "Failure" OR "Cure" OR "Outcome" OR "Partial response" OR "Dropout" OR "Treatment extension" OR "Resistance". In addition, the following key terms were used to search in ScienceDirect: "Localized Cutaneous Leishmaniasis" OR "Mucocutaneous Leishmaniasis" OR "Diffuse Cutaneous Leishmaniasis" OR "Old World Cutaneous Leishmaniasis" OR "*Leishmania aethiopica*" AND "Treatment relapse" OR "Treatment failure" OR "Treatment Cure" OR "Partial response". The detailed search strategy for the major databases is available in the S1 Text. To access grey literature, we navigated the first 20 hits in

Google Scholar using the review title. Moreover, manual search of the reference list of articles was done.

## Study selection process and quality assessment

To remove duplicates, we imported all studies retrieved from the databases and grey literature searches into Endnote X8. Two independent reviewers (AYA and KA) did the screening and selection of studies based on the title and abstract. These two reviewers did the critical appraisal of full-text articles. The authors use the Newcastle Ottawa Scale (NOS) [21], and the risk of bias assessment tool version 2 (RoB2) [22] to appraise observational and clinical trial studies, respectively. We presented the NOS result of the observational studies in Table A in S1 Table, and the RoB2 result in Table B in S1 Table. The authors resolved all differences during the appraisal process through discussion. The PRISMA 2020 flow diagram was followed to present the study selection process [20].

## Data extraction and outcomes

To extract the data, we developed a Microsoft Excel sheet based on the PRISMA template, and this template was piloted before use. AYA and KA extracted the following data items: 1) primary author and year of publication; 2) study design; 3) diagnosis confirmation method; 4) sample size; 5) clinical phenotype of CL (LCL, MCL and DCL); 6) treatment detail (route, dose, dosage, and treatment type); 7) characteristics of participants; and 8) treatment outcomes. The results for these data items are available in Table A to C in S2 Table. We classified treatment outcomes as primary and secondary. While favourable outcomes were considered primary, unfavourable outcomes were considered secondary.

## Data analysis

We did the narrative synthesis of evidence. In addition, a random effects model meta-analysis was done using a *metaprop* package in Stata 17. To include proportions close to 0% or 100% in the meta-analysis, we did the Freman-Tukey Transformation (ftt) of proportions using a *metaprop* package [23]. The pooled proportions of cure, partial response, relapse, dropout, and unresponsiveness reported by two or more studies were estimated and presented using forest plots or tables. We assessed heterogeneity for I-squared statistics <75% and non-overlapping confidence intervals on the forest plot [24]. To address high heterogeneity, we did subgroup and sensitivity analysis. Subgroup analysis was done by study design, clinical phenotype of CL, and type of treatment. To assess the small study effect, we did a subjective assessment of the funnel plot and an objective evaluation of Egger's test.

# Results

## Search results

Our comprehensive search retrieved 3520 studies from all sources. Of these, we selected 52 studies pertinent to our objective. Twenty-two studies reporting treatment outcomes of CL due to *L. aethiopica* were included in the narrative review. Finally, seven studies were included in the meta-analysis (Fig 1).

## Description and quality of included studies

A total of 808 study subjects participated in the 22 studies [8–11, 13–18, 25–36]. Twelve studies (nine case series [8, 9, 25–27, 29, 30, 32, 33] and three case reports [10, 11, 35]) documented treatment outcomes as part of routine clinical practice. All case series and case reports scored

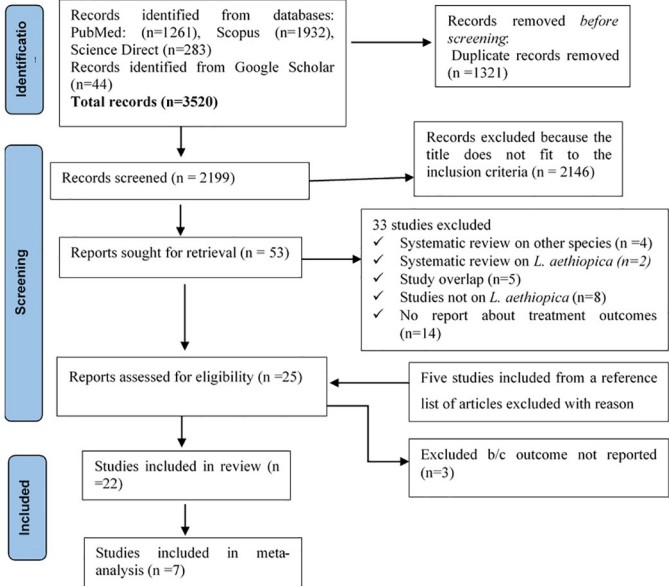

**Fig 1. The PRISMA flow diagram.**

less than five in the NOS, but seven observational studies (five cohort studies [14, 15, 17, 18, 34] and two retrospective chart reviews [16, 36]) scored five and above (Table A in S1 Table). As shown in Table B in S1 Table, three studies were clinical trials [13, 28, 31], but only one has low RoB [13]. Our review included 18 studies from Ethiopia [13–18, 25–36], three studies done among migrants and travelers [9–11], and one study from Kenya [8]. Twelve studies (nine case series and three case reports) [8–11, 25–27, 29, 30, 32, 33, 35] involve 65 study participants. Sixty-six participants were involved in three clinical trial studies [13, 28, 31]. In the seven studies [14–18, 34, 36] which were used for meta-analysis, 677 study subjects participated.

In the 16 studies [8–10, 13–18, 25, 26, 28, 32–36], the authors did parasitological confirmation of CL using SSS. The clinical phenotypes of CL were reported in 17 studies [11, 13, 14, 16–18, 25, 27–36] and unspecified in five studies [8–10, 15, 26]. Ten studies [11, 13, 14, 16–18, 31, 33, 34, 36] documented LCL clinical phenotype. Most studies recorded treatment outcomes with antimonial drugs. We present the details of study designs, sample size, confirmation of diagnosis, treatment type, dose, and dosage, patient characteristics, and responses to anti-leishmaniasis therapy in the S2 Table.

## Treatment outcomes

The primary studies included in this review documented favourable treatment outcomes such as cure, improved, positive outcome, negative/decrease in parasite density, good response, decrease in the lesion size, and resolution of the lesion without a scar. Unfavourable outcomes recorded in the studies were recurrence, resistance, treatment extension, partial response, relapse, dropout, unresponsiveness, progression to a more severe stage, and toxicity. The authors of the primary studies also documented the predictors of treatment outcomes.

**Favourable outcomes.** As shown in Table A in S2 Table, six Ethiopian cases with concomitant lesions (CL and leprosy) treated with systemic pentamidine responded well, but the authors did not report the route, dose, and duration of treatment, as well as outcome

ascertainment methods [26]. A prospective cohort study recorded a favourable response as a positive outcome and revealed that IM-Meglumine antimonate (MA) resistant cases treated with 4 mg/kg IM pentamidine every other day (QoD) for 20 days had shown clinical and microscopy cure at six months [14]. Another cohort study documented a good clinical improvement in 70% of cases at day 28, 28% at day 90, and 14.3% at day 180 follow-up, with a 100–150 mg daily oral miltefosine treatment [17]. Another cohort study noted that 14% of LCL patients treated with six doses of IL-SSG[V] have shown good response at day 90 [18]. As shown in Table A in S2 Table, five Kenyans treated with a high (18–20 mg/kg twice daily) dose of IV-SSG[V] were culture and smear-negative at 14–27 days and 3–18 months follow-up [8]. Three studies [9–11] also report treatment outcomes among travelers and immigrant cases. A Belgian traveler who returned from Ethiopia treated with 1.5 mg/5ml IM-MA for 45 days had clinical resolution at 45 days, and no relapse at six months [11]. An immigrant from Eretria to Germany treated with 200 mg/day IV-amphotericin B for 22 days had healing of the lesion without recurrence at 12 months of follow-up [10]. Three Ethiopian migrants in Israel had shown parasite clearance (smear and culture negative), clinical improvement and complete healing without a scar at 10 days, 20 days, and 40 days with a combination of 15% paromomycin sulfate and 12% methyl benzethonium chloride ointment [9].

Three clinical trials done in Ethiopia reported favourable treatment outcomes (Table C in S2 Table). A phase II clinical trial compared the efficacy of four weeks twice daily Shiunko ointment with a placebo among 20 LCL cases in each group. Five cases in the Shiunko group and four in the placebo group showed a cure. Six cases had a partial response to the Shiunko and placebo. While the Shiunko group showed a 69% reduction in lesion size, only 22% in the placebo group at 16 weeks [13]. Van der Meulen et al. [28] compared treatment outcomes in twelve CL cases. The authors administered a daily dose of three (isoniazid 300 mg, amithiozone 150 mg, and rifampicin 600 mg) combination drugs for eight weeks in the treatment group, and treated the control group with fifteen doses of 4 mg/kg IM pentamidine QoD. One case improved without inflammation and scar at 12 weeks in the treatment group, and six cases in the pentamidine group had negative smears at four weeks. In this trial, the authors did not report the clinical phenotype of CL. Another clinical trial compared a 200 mg itraconazole daily dose with a placebo in 14 patients (four DCL and 10 LCL cases), seven in each group [31], and documented no difference between the cure rate in the two groups after four weeks.

Four case series studies [25, 29, 30, 32] documented positive treatment outcomes among DCL cases. As shown in Table A in S2 Table, among 33 cases, seven were cured with pentamidine, three with chloroquine, one with primaquine and one with Amphotericin B [25]. Yet, the authors did not report the dates of outcome ascertainment. In another study [29], two DCL cases treated with 4 mg/kg IM pentamidine daily for 2 weeks showed a decrease in parasite number. This study authors indicated negative smear microscopy and biopsy as tests of cure, but they did not document the date of outcome ascertainment. Similarly, three DCL cases treated with 1 g 25% Chlorpromazine and 25% 50 g Vaseline ointment for one month showed the disappearance of clinical signs of inflammation in one case and decreased lesion size in another [30]. Teklemariam et al. [32] documented treatment outcomes of two cases treated with 14 mg/kg daily IM-Aminosidine for 60 days and one case treated with a combination of this same dose of Aminosidine plus 10 mg/kg daily IM-SSG. These authors report that all cases had cures after two months of extended treatment.

**Unfavourable outcomes.** In Ethiopia, two CL-HIV co-infected cases treated with 20 mg/kg/day IV-SSG[V] for 30 days had recurrence at five months of follow-up [33]. A case of MCL progressed to DCL after 30 days of treatment with a combination of 20 mg/kg/day IM-SSG[V] and 15 mg/kg/day paromomycin, and IM-SSG[V] alone for an extra 60 days [35]. In five MCL cases treated with four to eight weeks of 500 mg oral metronidazole, no changes were seen,

and two of the cases reported a burning sensation [27]. As shown in Table A in S2 Table, in a case series of 33 DCL patients [25], the authors documented that more than half of the participants were unresponsive to pentamidine and pentostam, and the study noted high toxicity (3 +) recorded with pentamidine and amphotericin B treatment. In a cohort study, while five LCL cases got worse their lesion, two cases were unresponsive to IL-SSG$^V$ [18].

Few clinical trial studies also reported unfavourable outcomes (Table C in S2 Table). In a phase II clinical trial that compared Shiunko ointment versus placebo, among the 20 LCL cases, six cases had partial response, eight had treatment failure and four dropped the treatment [13]. Similarly, a study comparing a four-week 50 mg oral itraconazole and placebo recorded active CL lesions in five of the seven cases at the follow-up visits [31]. In a study that compared a combination of 300 mg isoniazid, 150 mg amithiozone and 600 mg rifampicin with pentamidine, five of the six cases showed parasitological or clinical improvement after eight weeks of daily treatment [28].

**Predictors of outcome.** In this systematic review, we found two cohort studies in Ethiopia reporting predictors of CL treatment outcomes [17, 18]. A study by Tilahun et al. [18] noted that older age, SSS grade +1 and +2 and being male have an association with a high probability of cure at day 90 in LCL cases treated with IL-SSG$^V$. In a cohort study that assessed the outcome of patients treated with 150 mg/day oral miltefosine for one month, MCL predicted a low chance of relapse at day 180 compared to LCL [17]. In both studies, large lesion size decreases the chance of cure at day 90 [17, 18].

## Meta-analysis

Seven observational studies done in Ethiopia [14–18, 34, 36] were included in the meta-analysis. Although the subjective judgment of the funnel plot showed asymmetry (Fig 2), insignificant Egger's test (p-value = 0.329) indicates no publication.

**Proportion of cure.** The pooled proportion of cure estimated from seven studies [14–18, 34, 36] was 0.63 (95% CI; 0.38–0.86; $I^2$ = 97.81%) (Fig 3). This pooled estimate did not include the cure rate after treatment extension, retreatment or change in treatment regimen. The primary studies report the proportions of cure rate ranging from 55–95% at day 90 [17, 18, 34,

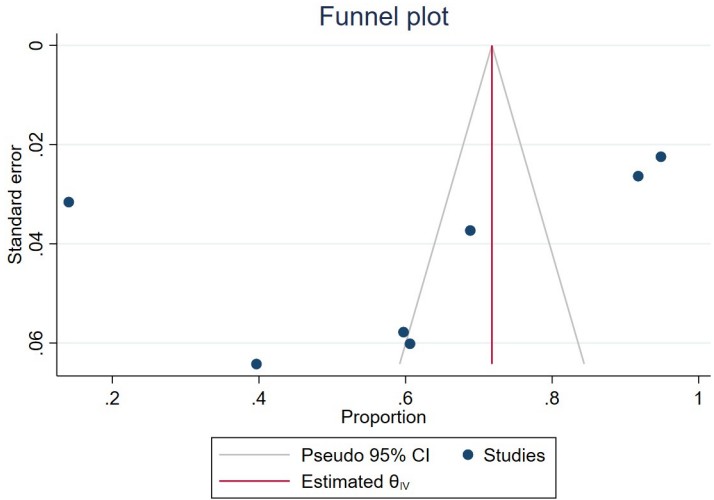

**Fig 2. Funnel plot.**

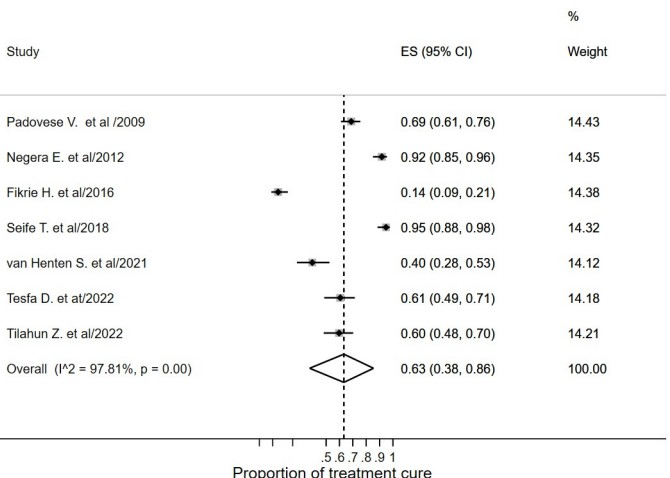

**Fig 3. Forest plot of the pooled proportion of CL treatment cure.**

36], and 62–69% at day 180 [14, 17]. A cohort study by van Henten S. et al. [17] reported an 18.2% cure rate at day 28.

*Subgroup analysis.* The I-square was high (97.81%) in the overall pooled estimate of the cure rate. Hence, the subgroup analysis was performed by study design, treatment type and clinical phenotype. Subgroup analysis by study design showed a 0.74 (95% CI: 0.57–0.91; $I^2$ = 95.8%) cure rate in the cohort studies, and 0.29 (95% CI: 0.22–0.35; $I^2 = 0$) in the retrospective chart reviews (Fig 4). The I-square remained high after subgroup analysis.

As shown in Fig 5, the subgroup analysis pooled proportion of cure with systemic antimonial treatment was 0.61 (95% CI: 0.29–0.89; $I^2$ = 96.68%), and 0.87 (95% CI: 0.67–0.99; $I^2$ = 91.41%) with local therapy. The studies documented systemic SSG[V] for MCL, DCL, and

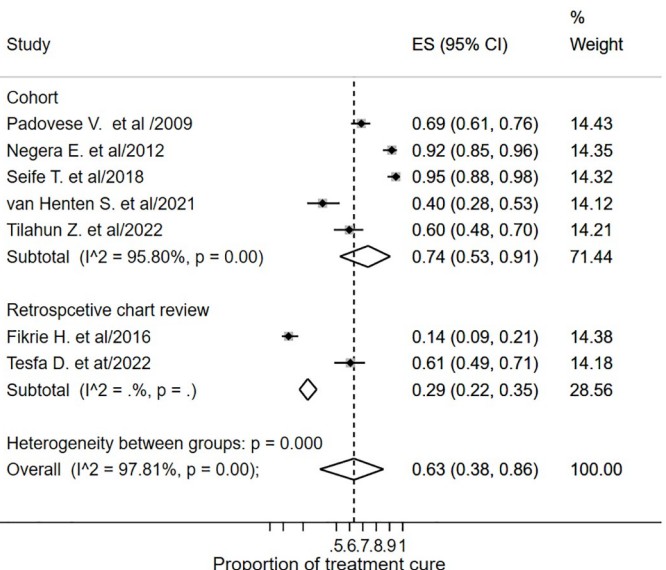

**Fig 4. Subgroup analysis forest plot of treatment cure by study design.**

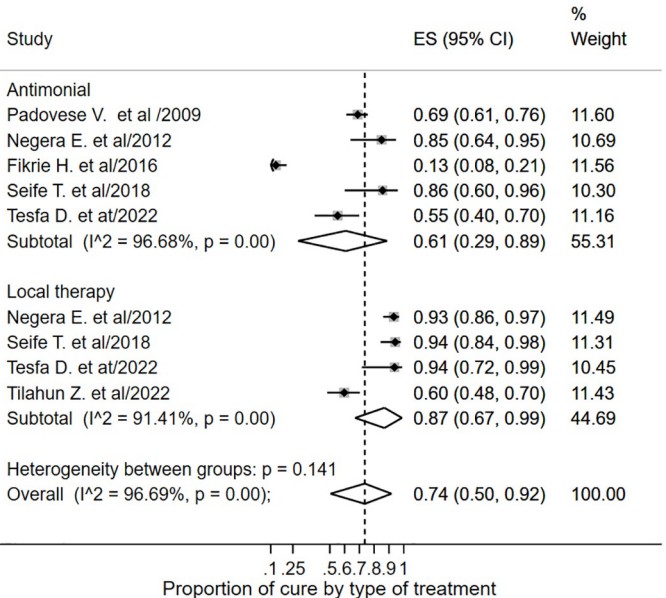

**Fig 5. Subgroup analysis forest plot of treatment cure by treatment type.**

unresponsive LCL treatment, as well as local therapy for the LCL and satellite MCL and DCL lesions. The types of local therapy reported were IL-SSG[V] [18, 36], cryotherapy [15] and a combination of IL-SSG[V] and cryotherapy [34]. In addition, a cohort study by Seife et al. [34] reported a cure rate of 94.8% with combination therapy for the clinical phenotypes MCL, DCL, and LCL. These authors report cure rates for MCL and DCL cases with a combination of IM-SSGV and allopurinol treatment, while documenting outcomes for LCL cases with IL-SSGV plus cryotherapy.

The subgroup analysis across LCL, MCL and DCL showed five studies [16–18, 34, 36] reporting cure rate for LCL cases, and the pooled proportion was 0.54 (95% CI: 0.18–0.88; $I^2$ = 96.73%) (Fig 6). While two studies [16, 36] documented IM-SSG[V] for LCL treatment, IL-SSG[V] was recorded in the other [18]. Fikre et al. [16] document unknown treatment outcomes in 35 LCL cases treated with IM-SSG[V]. In a cohort study, LCL cases showed a 94% overall cure rate [34]. This proportion was 92.3% with cryotherapy alone, and 96% with a combination of IL-SSG[V] and cryotherapy. In another cohort study, four out of twelve (19%) LCL cases showed cure with oral miltefosine [17].

The pooled proportion of cure was 0.49 (95% CI: 0.21–0.77; $I^2$ = 91.68%) in MCL cases (Fig 6). None of the studies [16, 17, 34, 36] documented cryotherapy in MCL treatment. In two studies, [16, 36] the proportion of cure ranged from 19–55% with IM-SSG[V] treatment, and one study [16] records unknown outcomes in 23 cases. While 79% of the MCL cases treated with a combination of IM-SSG[V] and Allopurinol, and 85.7% treated with a combination of IM-SSG[V] and IL-SSG[V] had shown cure, the overall MCL cure rate was 82% [34].

As depicted in Fig 6, the proportion of cure among DCL cases was 0.51 (95% CI: 0.23–0.79; $I^2$ = 71.74%). Three studies [16, 17, 36] report a below 50% cure rate in DCL cases with parenteral SSG[V] or oral miltefosine monotherapy. A cohort study [34] found that a combination of IM-SSGV and Allopurinol cured 80% of DCL cases, while a combination of IM-SSGV, cryotherapy, and IL-SSG had an 86% success rate. Cryotherapy and IL-SSGV were effective for DCL satellite lesions.

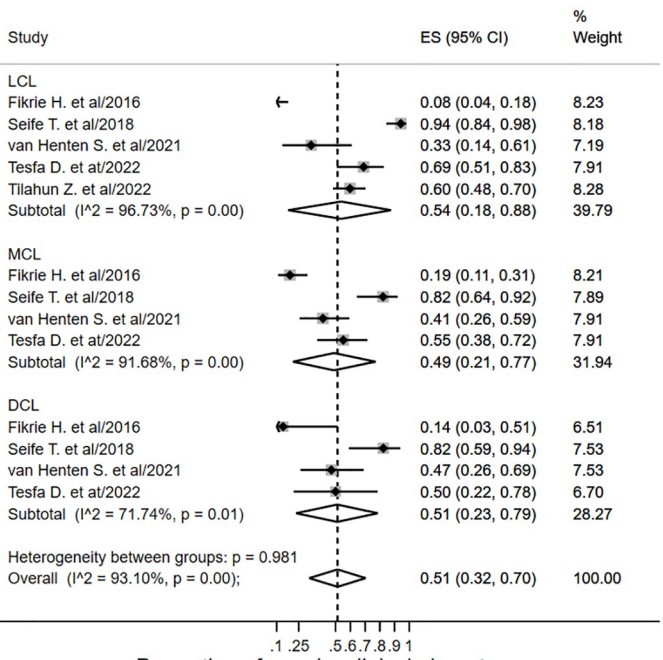

**Fig 6. Subgroup analysis forest plot of treatment cure by clinical phenotype.**

*Sensitivity analysis*. In the sensitivity analysis, no study affected the overall pooled proportion of treatment cure, but a study by van Henten et al. [17] showed some level of influence on the overall cure rate (Fig 7).

**Proportions of unfavourable outcomes.** We noted unfavourable outcomes in a cohort study [14]. The authors documented a 15% negative outcome among cases treated with 20 mg/kg/day IL-MA every 3 days for four weeks, and a 28% resistance with 20 mg/kg/day IV-MA four weeks treatment. The study also showed that 52% of relapsed cases retreated with the above same dose of IV-MA for four weeks did not show treatment cure. Another cohort study by van Henten S. et al [17] showed a 32.3% relapse at day 180 with four weeks of 50 mg daily oral miltefosine treatment. While van Henten S. et al [17] noted a 20.4% partial response, a study by Fikre et al. [16] documented a 25% partial response. Table B in the S2 Table showed the detail treatment outcomes at different time points. Moreover, we estimated the pooled proportion of unfavourable treatment outcomes (partial response, relapse, dropout, and unresponsiveness) reported by two or more studies. In these estimations, all I-square statistics results were below 75% (Table 1).

## Discussion

In this systematic review, we summarized evidence on favourable and unfavourable treatment outcomes of CL due to *L. aethiopica*. Primary studies documented systemic antimonial monotherapy, local (topical) or combination therapies. The pooled proportion of cure was low. Particularly, systemic antimonial monotherapy showed a low cure rate in all clinical phenotypes (LCL, MCL and DCL). The reviewed studies report LCL had a higher cure with local therapy, and all clinical phenotypes demonstrated a better cure with combination therapy. Our review also found a significant proportion of unfavourable outcomes, such as partial response, relapse, dropout, and unresponsiveness.

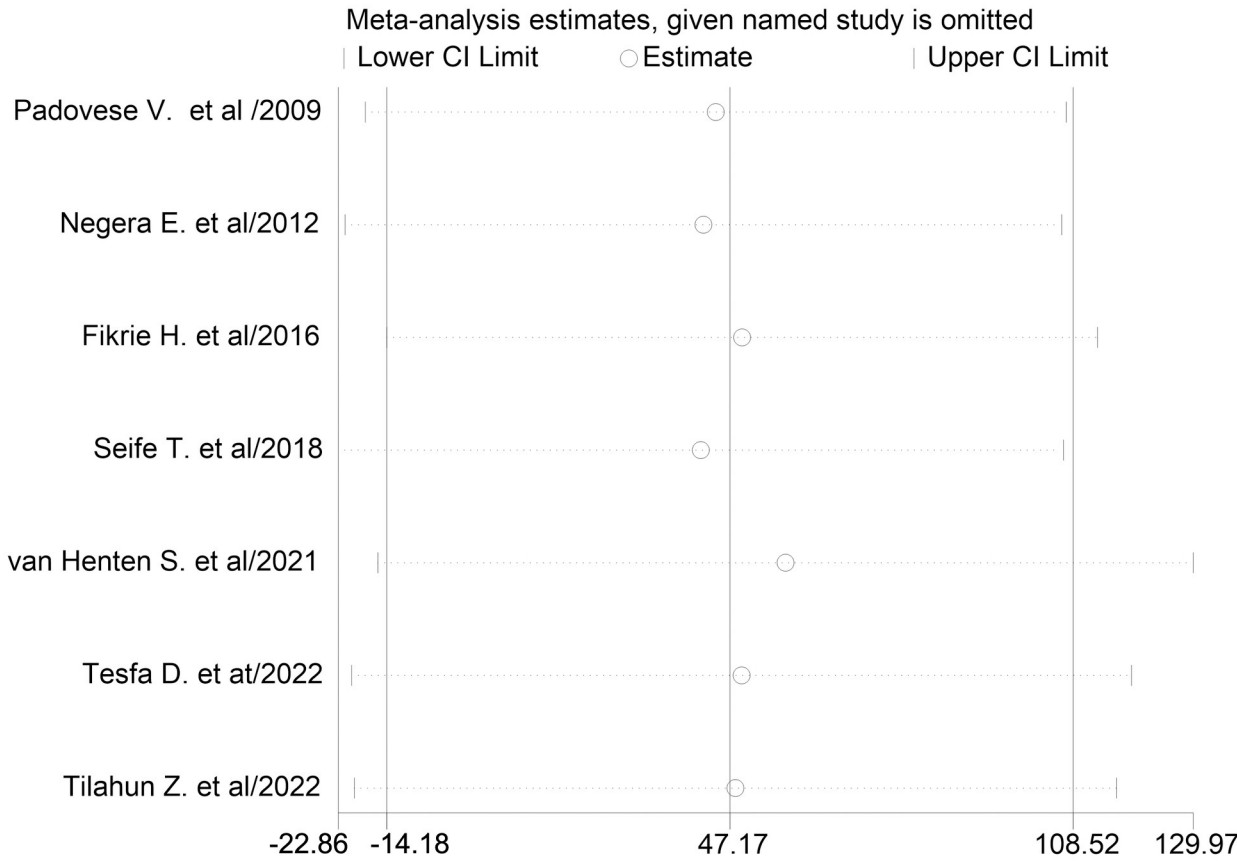

**Fig 7. Sensitivity analysis.**

In this study, the overall pooled proportion of cure was 63%. This proportion was low with systemic antimonial monotherapy, such as IV/IM-SSG$^V$. In this review, the subgroup analysis by type of treatment also showed the proportion of cure as low as 29% with systemic antimonial monotherapy, and an overall cure rate was 61%. This lower proportion might be due to the low efficacy of antimonial drugs and high drug resistance in *L. aethiopica*. A cohort study evaluating treatment outcomes of a presumably *L. aethiopica*-associated CL in our review documented 28% resistance to systemic meglumine antimonate [14]. A systematic review also reports an increasing resistance of *Leishmania* species to SSG$^V$ drugs [5]. Although no evaluation study done so far, an Ethiopian leishmaniasis diagnosis and treatment guideline also report the ineffectiveness of SSG in treating MCL and DCL due to *L. aethiopica* [12]. In our finding, despite less effectiveness and resistance, SSG$^V$ remains the mainstay of treatment. These necessities further study to assess the resistance pattern of this species for SSG$^v$, and

**Table 1. The pooled proportion of unfavourable outcomes.**

| Type of unfavourable outcome | Pooled proportion (95% CI) | I-squared statistics |
|---|---|---|
| Partial response [16–18, 36] | 0.19 (95% CI: 0.14–0.25) | 40.71% |
| Relapse [14, 17] | 0.17 (95% CI: 0.12–0.22) | 0 |
| Dropout [15, 17] | 0.19 (95% CI: 0.13–0.25) | 0 |
| Unresponsiveness [15, 16, 18] | 0.06 (95% CI: 0.03–0.09) | 41% |

update the national CL treatment guideline in Ethiopia with the inclusion of other treatment options.

A systematic review reports a 70–100% cure rate with a combination of pentavalent antimonial with other alternatives [38], but without reporting leishmania species and the clinical phenotypes. In our review, only one cohort study documented treatment outcomes with various combination therapies [34]. This cohort study revealed that a combination of SSG$^V$ with other treatment options has a high cure rate ranging from 78.6–96% across various clinical phenotypes. Specifically, the authors report a 96% cure rate in LCL cases with a combination of IL-SSG$^V$ and cryotherapy. In MCL cases, the proportion was 79% with a combination of IM-SSG$^V$ and IL-SSG$^V$, and 85.7% with a combination of IM-SSG$^V$ and Allopurinol. In DCL cases, the proportion of cure was 80% with a combination of IM-SSG$^V$ and Allopurinol, and 86% with a combination of IM-SSG$^V$, cryotherapy and IL-SSG$^V$. Hence, combination therapies are the potential choice for clinicians treating CL caused by *L. aethiopica*, but high-quality randomized control trials are essential to evaluate the effectiveness and safety of these therapies.

In the subgroup analysis, CL due to *L. aethiopica* showed an 87% cure rate with local therapy. This finding is in line with a systematic review [37] that reports an 80% cure rate for CL due to *Leishmania. major* and *L. tropica* with topical agents. Various local therapeutic options such as thermotherapy, paromomycin, and combination, $CO_2$ laser, 5-aminolevulinic acid hydrochloride (10%) plus visible red light (633 nm) and cryotherapy are available to treat LCL in other settings [37]. However, our review documented only liquid nitrogen-based cryotherapy and IL-SSG [15, 18, 34, 36]. In all the studies reviewed, IL-SSG was the main topical agent, and the effectiveness of this topical agent alone was low. For example, in a cohort study we reviewed, the cure rate was 60% with IL-SSG [18]. In addition, liquid nitrogen-based cryotherapy is expensive at an estimated cost of 4 USD per patient [15]. Hence, other topical agents such as thermotherapy and $CO_2$ laser are potentially effective treatment options for CL due to *L. aethiopica*, but the effectiveness of these therapies requires further study.

In this study, we found no significant difference in cure rate across the different clinical phenotypes. A cohort study also showed no significant difference in cure rate between DCL and LCL cases [17]. Because most primary studies included in our review assessed outcomes based on the routine clinical data collected in the referral settings, these studies probably included complicated CL cases that did not exhibit different probabilities of treatment cure. However, in our review, the combination of SSG$^V$ with other treatment options showed a high cure rate ranging from 78.6–96% across various clinical phenotypes [34]. The proportion of cure was 96% in LCL cases, with a combination of IL-SSG$^V$ and cryotherapy. In MCL cases, the proportion was 79% with a combination of IM-SSG$^V$ and IL-SSG$^V$, and 85.7% with a combination of IM-SSG$^V$ and Allopurinol. In DCL cases, the proportion of cure was 80% with a combination of IM-SSG$^V$ and Allopurinol, and 86% with a combination of IM-SSG$^V$, cryotherapy and IL-SSG$^V$ [34]. A systematic review also reports a 70–100% cure rate with a combination of pentavalent antimonial with other alternatives [38], but without reporting leishmania species and the clinical phenotypes. Although clinicians may consider combination therapies for CL due to *L. aethiopica*, high-quality randomized control trials are essential to evaluate the effectiveness and safety of combination therapy.

In this study, the findings showed the pooled proportion of unfavourable outcomes, including partial response (19%), relapse (17%), dropout (19%) and unresponsiveness (6%). The proportion of relapse in our finding was higher than the 2–6% relapse reported in a systematic review that assessed the effectiveness of different topical agents for LCL cases due to *L. major* and *L. tropica* [37]. The high proportion of relapse in our finding might be due to severe clinical phenotypes such as MCL and DCL associated with *L. aethiopica* [3]. These severe clinical

phenotypes are difficult to treat [39]. Similarly, partial response and unresponsiveness were unacceptably high in our findings. This might be due to the high resistance of *L. aethiopica* to available treatment options; as described earlier. The dropout might also increase the probability of partial response, relapse, and unresponsiveness. Hence, due account for drug resistance and reducing dropout are vital to counter unfavourable outcomes.

This systematic review and meta-analysis provided updated and summarized evidence on treatment outcomes of CL due to *L. aethiopica* treated with available anti-leishmaniasis therapy. However, the study was not without limitations. The first limitation was heterogeneity persisted in the random effects model and subgroup analysis, but decision-makers could use the summary estimates with cautious interpretation, as evidence from two studies is more informative for decisions than evidence from one clinical trial [40]. Second, the number of studies in the meta-analysis was below ten, which might affect the reliability of the funnel plot and Egger's test. Furthermore, clinical trial studies included in this study had low quality, involved small sample size, and none of the studies assessed the effectiveness of either local, combination, or systemic antimonial therapy using high-quality randomized control trials.

## Conclusions

The pooled proportion of cure of CL caused by *L. aethiopica* was low. All clinical phenotypes showed a low cure rate with systemic antimonial monotherapy, but the localized type also showed better cure with local therapy. Combination therapies had a better cure rate for all clinical phenotypes, but a single study documented this evidence. The proportion of unfavourable treatment outcomes, partial response, relapse, dropout and unresponsiveness, were significant. Combination therapies are the potential choice of treatment for all clinical phenotypes. The primary studies included in this review assessed treatment outcomes based on observational or low-quality clinical trial studies. Future efforts should focus on searching for effective monotherapy and evaluating the effectiveness of combination therapy using high-quality randomized control trials.

## Supporting information

**S1 Table. Quality of studies included in the systematic review and meta-analysis.**
(DOCX)

**S2 Table. Characteristics of studies, treatment details, participant characteristics and treatment outcomes.**
(DOCX)

**S1 Checklist. PRISMA checklist.**
(DOCX)

**S1 Text. Searching strategy.**
(DOCX)

## Author Contributions

**Conceptualization:** Abebaw Yeshambel Alemu.

**Data curation:** Abebaw Yeshambel Alemu, Kassahun Alemu.

**Formal analysis:** Abebaw Yeshambel Alemu.

**Investigation:** Abebaw Yeshambel Alemu.

**Methodology:** Abebaw Yeshambel Alemu, Lemma Derseh, Mirgissa Kaba, Endalamaw Gadisa, Kassahun Alemu.

**Software:** Abebaw Yeshambel Alemu.

**Supervision:** Lemma Derseh, Mirgissa Kaba, Endalamaw Gadisa, Kassahun Alemu.

**Validation:** Endalamaw Gadisa.

**Visualization:** Mirgissa Kaba, Kassahun Alemu.

**Writing – original draft:** Abebaw Yeshambel Alemu.

**Writing – review & editing:** Abebaw Yeshambel Alemu, Lemma Derseh, Endalamaw Gadisa, Kassahun Alemu.

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
