## [Decision Letter · Decision Letter 0]

12 Mar 2023

PONE-D-23-03546Treatment outcomes of cutaneous leishmaniasis due to Leishmania aethiopica: A systematic review and meta-analysisPLOS ONE

Dear Dr. Alemu,

Thank you for submitting your manuscript to PLOS ONE. After careful consideration, we feel that it has merit but does not fully meet PLOS ONE’s publication criteria as it currently stands. Therefore, we invite you to submit a revised version of the manuscript that addresses the points raised during the review process.

We look forward to receiving your revised manuscript.

Kind regards,

Balew Arega Negatie, Msc,MD

Academic Editor

PLOS ONE

Journal Requirements:

**Additional Editor Comments:**

See my comments  attached in main text tracked.

Reviewers' comments:

Reviewer's Responses to Questions

**Comments to the Author**

1. Is the manuscript technically sound, and do the data support the conclusions?

Reviewer #1: No

Reviewer #2: Yes

2. Has the statistical analysis been performed appropriately and rigorously? 

Reviewer #1: No

Reviewer #2: Yes

3. Have the authors made all data underlying the findings in their manuscript fully available?

Reviewer #1: Yes

Reviewer #2: Yes

4. Is the manuscript presented in an intelligible fashion and written in standard English?

Reviewer #1: Yes

Reviewer #2: Yes

5. Review Comments to the Author

Reviewer #1: the title itself done previously, I found a lot of errors particularly in the methodology section, topographical errors and also I did not found new compared to the published (Major revision required).

Reviewer #2: I have got the manuscript written well. It brings the important agenda to the board, the treatment outcomes of CL.

However, I do have some comments to be considered:

1. I include the review/publication period in the abstract and method section. E.g. in the review, articles published between 2009-2022 were included

2. Geographical location of the study. Ethiopia? East Africa? Limit its boundary. 18 studies are from Ethiopia, 3 from migrants and 1 in Kenya. This is good to infer the study findings.

3. Abbreviation NOS (Non-observational studies?) not clear. Clarify it (page 6)

4. Your suggestion and recommendation regarding local therapy for CL due to L. aethopica.

6. PLOS authors have the option to publish the peer review history of their article (what does this mean?). If published, this will include your full peer review and any attached files.

Reviewer #1: No

Reviewer #2: **Yes: **Abdella Gemechu

---

## [Author Response · Author response to Decision Letter 0]

17 Apr 2023

Point by a point response letter

Dear Academic Editor (Dr. Balew Arega Negatie, MD, MSc), Reviewer #1, and Reviewer #2

After going through the entire manuscript, you forwarded your constructive comments which we missed to touch. Therefore, we are glad enough to express our sincerest thanks for your constructive editorial and review comments that could help to improve the novelty of our effort. We first presented the comments and author response to the editor comments followed by Reviewer #1 and Reviewer #2, respectively. The comments were highlighted yellow and the author response highlighted pink. 

Point by point response to academic editor’s comment

Editor’s general comment 

Comment 1. Please ensure that your manuscript meets PLOS ONE's style requirements, including those for file naming. The PLOS ONE style templates can be found at 

Response: Thank you dear reviewer for your honorable comment. We followed the PLOS ONE author’s guideline to prepare the manuscript. The figures were prepared using PACE and uploaded separately. Additionally, we prepared the tables and supporting files as per the authors guideline. 

Additional Editor comments taken from tacked manuscript

Comment: This is good if rephase like this “Leishmania aethiopica is a unique species that causes cutaneous leishmaniasis, and studies evaluating the effectiveness of treatment for this condition reported inconsistent findings

Response: Thank you dear editor for the suggestion. It has been revised as per the comment and the changes made are available in track change in the abstract section. 

Comment: Change to “decisions “

Response: Thank you so much. It has been edited as per the comment in the revised document. 

Comment: The narrative

Response: Thank you editor. We revised the grammar as per the comment. 

Comment: Remove this sentences

Response: Thank you dear editor once again. The statement has been removed from the abstract section as per the comment and the change can be seen in the track change. 

Comment: The conclusion is based on the single study finding which is not appropriate

Response: Dear editor you for the comment. We made revision as per the comment and the changes can be seen in track change in the conclusion section of the abstract. 

Comment: This can be rephrased as “Compared to antimonial monotherapy, combination therapy has a better clinical phenotype cure rate”

Response: Thank you for the suggestion. We revised the conclusion as per the suggestion and comment. The revisions made can be seen in the conclusion section of the abstract. 

Comment: Was reported

Response: Thank you once again. We revised as per the comment and the changes can be seen in track changes in the revised manuscript. 

Comment: This paragraph is not related to the different species of Leishmania. So, either remove it totally or make it short.

Response: Thank you for your comment. We make the paragraph short than removing it. We did not remove it because the responses to anti-leishmaniasis therapy were our focus. So the existing knowledge about the difference in species level response to treatment was summarized. The response of L.tropica and L.major to different treatment was summarized. 

Comment: Here, please writes the guideline-recommended treatment regimen in the country if any.

Response: Thank you dear editor for the comment. 

Comment: “Summarizes” is a better word than the aggregate

Response: Thank you dear editor for your suggestion. We revised it accordingly and the changes can be seen in the introduction section at page 

Comment: Why do you ignore the heterogenicity? How do you do MA while there is significant heterogenicity?

Response: Thank you dear editor. Yes, there were methodological and clinical heterogeneity between the studies that we reviewed, but this did not limit authors to perform meta-analysis [1]. Hence, we did met-analysis and checked statistical heterogeneity. Because we detected high I-square >75% we did subgroup analysis and sensitivity analysis and report these results with the main analysis for the readers. It is possible to conduct meta-analysis despite the methodological and clinical heterogeneity, and when statistical heterogeneity exists, we can do subgroup analysis based on the clinical and methodological variation. 

Comment: See above comment, you already plan to ignore heterogenicity, what is the importance of this sentence and subsequent below?

Response: 

Comment: “Favorable vs unfavorable” is better wording than positive vs negative

Response: Thank you dear editor. We revised as per the comment and the changes made can be seen in the track change in the results section. 

Comment: In the presence of such high heterogenicity, what is the importance of determining the pooled estimate? It good simply narrate the finding in such a cases.

Response: Thank you dear editor. Yes of course combining apples and oranges is not advisable. Our interest was not to compare the effect of different treatment options rather we estimated the pooled estimates of response of L. aethiopica for anti-leishmaniasis therapy. When the researcher’s interest is to estimate the response of a disease for various treatment options, in general, it is possible to pool the effect size [2]. Furthermore, meta-analysis should be conducted when a group of studies is sufficiently homogeneous in terms of subjects involved, interventions, and outcomes to provide a meaningful summary. However, it is often appropriate to take a broader perspective in a meta-analysis than in a single clinical trial [3].

Comment: In Figure 5, the is only one study in the subgroup analysis (Seife T/2018). However, there must be at least two studies in each subgroup. The finding of this analysis skewed your conclusion and recommendation. This must be corrected and revise your conclusion accordingly.

Response: Thank you dear editor for your high-level curiosity and valid comment. Yes, there is only one study for the sub-group analysis by treatment type. The sub-group analysis was made using weighted random effects model. So, the result will not be skewed by Seife T et al /2018 because it accounts only for 10.27% in the sample weights. Since sub-group analysis is not primary analysis, this single study was isolated and weighted alone while the studies that report similar treatment type were grouped together. Besides, the pooled effect size in the sub-group analysis was not reported so that it would not skew the pooled estimate. Moreover, the narrative review report was presented together with the meta-analysis, so readers and decision makers can evaluate the finding. 

Comment: These sentences should be modified. In the presence of high heterogeneity, what is the importance of determining the pooled estimate?

Response: Thank you dear editor. Yes of course there is no recommendation to conduct meta-analysis based on a single study, but in the subgroup analysis only one study can be remained as single when those studies that showed similar grouping variable are grouped together. The study by Seife T et al /2018 remained as single when the subgroup of studies that report cure with antimonial therapy and local therapy were grouped together. This single study result can be seen in the forest plot, but weighted for the sample size. The study by Seife T et al /2018 had 10.27% contribution. 

Comment: Please support with a reference

Response: Thank you dear editor. We cited the reference as per your comment. 

Comment: Is there any recommendation to pool effect from a single study? see other comments too.

Response: Thank you dear editor. Your concern is highly respected. Yes of course there is no recommendation to conduct meta-analysis based on single study. The pooled effect of a single study shown in the forest plot of sub-group analysis by treatment was not primary meta-analysis result that aimed to pool the cure rate from a single study. 

Comment: See the comments about figure 5.

Response: Thank you indeed! The comment is highly valued. The case of Figure 5 was to show the contribution of combination therapy by Seife T et al /2018 in the sub-group analysis. While antimonial and local therapy incorporated more than one study, combination therapy remained with only one study. The conclusion was narrated based on the result from the review, and we did not include cure rate with percentage to avoid misleading the readers and decision makers. 

Point by point response to comments by Reviewer # 1 

General comments to the Author by reviewer #1

Comment 1. Is the manuscript technically sound, and do the data support the conclusions?

Reviewer #1: No

Response: Thank you dear reviewer. We revised the methodology for technical soundness of the study as per your comments. The revisions made can be depicted in track changes in the materials and methods section of the revised manuscript. 

Comment 2. Has the statistical analysis been performed appropriately and rigorously?

Reviewer #1: No

Response: Thank you dear reviewer. We are grateful for your scholarly comments. The manuscript has been revised intensively as per your comments. The changes made can be seen in track changes throughout the manuscript. 

Comment 3. Have the authors made all data underlying the findings in their manuscript fully available?

Reviewer #1: Yes

Response: Thank you dear reviewer for your encouraging comment. 

Comment 4. Is the manuscript presented in an intelligible fashion and written in standard English?

Reviewer #1: Yes

Response: Thank you dear reviewer for your encouraging comment. 

Specific review comments to the author by reviewer #1 taken from the attachment file 

Comment: The authors conducted a study on treatment outcomes of cutaneous leishmaniasis due to Leishmania aethiopica: A systematic review and meta-analysis. This is a very interesting and relevant study, but I suggest major changes. 

Response: Dear reviewer thank you for your encouraging comment and indicating points for improvement of our work. Major revisions have been made as per your comments and suggestions as highlighted with track changes in the manuscript. 

Title 

Comment: similar study was done in 2016 (Treatment of Cutaneous Leishmaniasis Caused by Leishmania aethiopica: A Systematic Review by Johan van Griensven, Endalamaw Gadisa, Abraham Aseffa, Asrat Hailu, Abate Mulugeta Beshah and Ermias Diro). So, what is new? 

Response: Thank you so much dear reviewer for your meticulous reading and comment. We accept your comment with great honor. However, our systematic review and meta-analysis generated an additional evidence to the existing review, and we generated the following new evidences to existing knowledge on the responses of L. aethiopica to anti-leishmaniasis therapy. 

1. We estimated the pooled proportion of cure based on the meta-analysis of proportions of cure reported from seven studies, this might be taken as additional evidence. However, the previous article did not report the pooled proportion of cure. 

2. The secondary outcomes of our systematic review and meta-analysis also revealed the pooled proportions of unfavorable outcomes, such as treatment failure, partial response, dropout and unresponsiveness. Nonetheless, the previous study did not pool evidences on these outcomes. We felt that these reports are also additional evidence to the scientific community to anticipate the proportions of unfavorable treatment outcomes while treating CL due to L. aethiopica. 

3. Unlike the previous narrative review, the pooled proportion of cure was reported based on the clinical phenotype of CL and type of treatment. We felt that this is also additional evidence to the existing body of knowledge. 

4. After systematic review done in 2016 eight studies published. We included these 8 new studies in our systematic review and meta-analysis that gave us the chance to pool the favorable and unfavorable responses of L. aethiopica to anti-leishmaniasis therapy, but his reports were impossible in the previous review because the studies were not enough to generate pooled evidence. 

5. It has been seven years since the previous systematic review has been published, so our study generated more recent evidence. Additionally, I consulted a senior Biostatician and an Associate professor at University of Gondar about the possibility of doing systematic review and meta-analysis in the presence of published systematic review. He affirmed that new systematic review and meta-analysis required after five years to generate additional knowledge, provide new studies are added. 

Comment: Author affiliations: check the affiliation of Author one (Debre Tabor and Gondar Universities?) 

Response: Thank you for your curiosity comment. Yes of course it is not possible to get employment at two public institutions according to the Ethiopian government civil service rule and regulation. However, first author is a PhD student affiliated to University of Gondar, sponsored by Armauer Hansen Research Institute and the employee of Debre Tabor University. According the PhD guideline used by University of Gondar, publication by a PhD student should have affiliation to University of Gondar so that the University’s contribution is recognized. The same works for Armauer Hansen Research Institute too. 

Introduction 

Comment: Reference 1: Cutaneous leishmaniasis is a category one emergent uncontrolled neglected tropical skin disease (skin NTD) caused by obligate intracellular protozoa; Leishmania [1]. Correct as Cutaneous leishmaniasis (CL)-------------and correct Leishmania as Genus leishmania 

Response: Heartful thanks for the suggestion. The revision has been made as can be seen in track changes in the revised document. 

Comment: It would be better to write the introduction from General to specific

Response: Thank you dear reviewer. We revised the manuscript as per the comment and the changes can be seen in the track changes in the revised manuscript. 

Comment: You mentioned Predictors of outcome in the result section. So please include information about predictors on the introduction section 

Response: Sure! We included the predictors of treatment outcome in the introduction section. The additions and changes made can be seen in track change in the fourth paragraph of the introduction section. 

Materials and Methods 

Eligibility criteria 

Comment: We included studies on human being that reported treatment outcomes of CL due to L. aethiopica, published in English but without restriction to year of publication and study design. Already included in the above-mentioned study, why do not you focus on the recent literatures and -------------- 

Response: Thank you dear reviewer. Yes, some of the studies were included in the previous systematic review, but those articles should not be removed because included in the past review. Besides one of the goals of conducting manual searching of the reference list of systematic reviews excluded is to include those studies included in that prior systematic review. 

Comment: What about case reports, narrative review with no full information, qualitative studies and soon???? 

Response: Thank you once again. We revised as per the comment, and we included the revision in the eligibility criteria sub-section of the methods and materials section. The changes made can be seen in track changes. 

Comment: Inclusion and exclusion criteria---Very shallow

Response: Thank you dear reviewer. We included the exclusion and inclusion criteria as per the comments and the changes made can be seen in the eligibility criteria sub-section. 

Comment: Data extraction and quality assessment: where is it?

Response: Thank you dear reviewer. We cited the supporting files in the methods section. The results of quality assessment for observation studies were presented in Table A in S1 Table, whilst the quality assessment result of clinical trial studies was presented in Table B in S1 Table. The data extraction results were presented from Table A to C in S2 Table in the results section. 

Comment: The following data items were extracted: 1) primary author and year of publication; 2) study design; 3) diagnosis confirmation method; 4) sample size; 5) clinical phenotype of CL (LCL, MCL and DCL); 6) treatment detail (route, dose, dosage and treatment type); 7) characteristics of participants; and 8) treatment outcomes. Where is your data that contains the above variables????

Response: Thank you dear reviewer. We cited the supporting file in the methods section and the results that contain data related to the above data items were presented in the results section from Table A to C in S2 Table. 

Results 

Comment: Characteristics and quality of included studies. It better to replace by Description of included studies and Characteristics of the included studies subtitles and state accordingly

Response: Thank you dear reviewer for your suggestion and valuable comment. We revised as per your comments. In addition to your suggestion, we added the subsection “search results” in the results section. The changes can be seen in the results section of the revised manuscript. 

Comment: Fig. 1 Flow chart of studies’ search and retrieval process should be presented in the main document

Response: Thank dear reviewer. We uploaded the figures separately because of the journal recommendation. The PLOS ONE authors guideline recommended that Figures should be uploaded separately and tables should be uploaded in the main document. 

Comment: In Fig.1, report sought retrieval (n=53) and of them 33 studies excluded =when we sum up the number of studies excluded are 19, 19 vs 33???? 

Response: Thank you dear reviewer for the highest level of curiosity. It was an editorial problem. The 14 studies excluded because responses to therapy were not reported were hidden during figure editorial process. So, we made revision to Figure 1 as per the comment. 

Comment: Five studies included from a reference list of articles excluded with reason. Why do you include those studies manually at the beginning in parallel with electronic searching engines

Response: Thank you dear reviewer. Your comments are highly valued. Five articles were included through manual searching of the reference list of excluded articles retrieved. These studies were retrieved from manual searching of the reference list of articles, but we were unable to put parallelly at the beginning because of the spacing to draw the figure. 

Comment: Where is your Subgroup analysis result

Response: Thank you dear reviewer. The results of the subgroup analysis were uploaded in Figure as separate TIF file. The Figures 4, 5 and 6 showed the subgroup analysis results. 

Comment: Clearly indicate trim and fill analysis for publication bias

Response: Thank you dear reviewer once again for your curiosity. The trim and fill analysis were not performed in our case. Despite the asymmetric funnel plot, the Egger’s regression p-value was 0.329, this implies no publication bias so we did not perform trim and fill analysis. 

Comment: What models you used to assess heterogeneity

Response: Thank so much. The I-square test was done to assess the heterogeneity. Random effects model was used to pool the proportion of cure in our study that gave the I-squared statistics that we used to evaluate heterogeneity between studies. 

Comment: Tables and figures should be presented in the result section of the main document

Response: Thank dear reviewer. Because of the rules of journal guideline, we were unable to present the figures in the main document. Additionally, the tables we used to present the data items was large to fit in single page border of the Microsoft word, so we included the main results table as supporting file in Tables A to C in S2 Table. All the data items and variables are presented in these supporting file tables. 

Comment: Show your Subgroup and sensitivity analysis result in the main document

Response: Thank you dear reviewer once again. The sensitivity analysis and subgroup analysis figure were uploaded separately as per the authors guideline recommendation to submit figures. The sub-group analysis result figures were uploaded as Figure Figures 4, 5 and 6 while Figure 7 showed sensitivity analysis result. We did not include the figures in the main document because of the journal recommendation that the figures should be uploaded separately based on their order as cited in the main document. 

Discussion 

Comment: Compare your finding with another systemic studies and or guidelines. If not available you can compare with studies with large sample size 

Response: Thank you dear reviewer. The references used for comparison in the discussion section were systematic reviews and meta-analysis. 

Comment: Please discus variables used in subgroup analysis like year… 

Response: Thank you reviewer for your comment. We included discussion on the subgroup analysis based on type of treatment and clinical phenotype of CL. We did not perform subgroup analysis by year, so we did not conduct discussion based on year. 

Comment: Discus on Predictors of outcome 

Response: Thank you dear reviewer. We revised as per the comment and the revision can be seen in track changes in the discussion section.

References 

Comment: Write your references following standard Vancouver referencing style 

Response: Thank you dear reviewer we revised as per the comments, and the changes can be seen 

Comment: In general: the title itself done previously, I found a lot of errors particularly in the methodology section, topographical errors and also I did not found new compared to the published.

Response 1: Thank you dear reviewer for your scholarly comment. The authors revised the methodology section in detail and the changes made can be seen in track changes in the methodology section of this manuscript. 

Response 2: Thank you dear reviewer. The typological errors are copy edited by the University of Gondar editorial center, and I contacted my colleague Mr. Yallew Aklog who is PhD fellow in English language at Bahir Dar University Ethiopia to review the grammar and copy edited the typological errors. 

Response 3: Thank you dear reviewer. The authors of this study are grateful for your valid comments, but our study generated the following new and additional evidence. 

1. We estimated the pooled proportion of cure based on the meta-analysis of proportions of cure reported from seven studies, this might be taken as additional evidence. However, the previous article did not report the pooled proportion of cure. 

2. The secondary outcomes of our systematic review and meta-analysis also revealed the pooled proportions of unfavorable outcomes, such as treatment failure, partial response, dropout and unresponsiveness. Nonetheless, the previous study did not pool evidences on these outcomes. We felt that these reports are also additional evidence to the scientific community to anticipate the proportions of unfavorable treatment outcomes while treating CL due to L. aethiopica. 

3. Unlike the previous narrative review, the pooled proportion of cure was reported based on the clinical phenotype of CL and type of treatment. We felt that this is also additional evidence to the existing body of knowledge. 

4. After systematic review done in 2016 eight studies published. We included these 8 new studies in our systematic review and meta-analysis that gave us the chance to pool the favorable and unfavorable outcomes of L. aethiopica to anti-leishmaniasis therapy, but these reports were not performed in the previous review because the studies were not enough to generate pooled effect size. 

5. It has been seven years since the previous systematic review has been published, so our study generated more recent evidence. Additionally, I consulted a senior Biostatician who is an Associate Professor at University of Gondar on the possibility of doing systematic review and meta-analysis in the presence of published systematic review. He affirmed that new systematic review and meta-analysis required after five years to generate additional knowledge, provide new studies are added. 

Point by point response to Reviewer #2 comments

General comments to the Author by reviewer #2

Comment 1. Is the manuscript technically sound, and do the data support the conclusions? 

Reviewer #2: Yes

Response: Thank you dear reviewer for your encouraging comment. 

Comment 2. Has the statistical analysis been performed appropriately and rigorously?

Reviewer #2: Yes

Response: Thank you dear reviewer for your encouraging comment

Comment 3. Have the authors made all data underlying the findings in their manuscript fully available?

Reviewer #2: Yes

Response: Thank you dear reviewer for your encouraging comment

Comment 4. Is the manuscript presented in an intelligible fashion and written in standard English?

Reviewer #2: Yes

Response: Thank you dear reviewer for your encouraging comment

Specific review comments to the author by reviewer #2

Comment: I have got the manuscript written well. It brings the important agenda to the board, the treatment outcomes of CL.

Response: Thank you dear reviewer. The authors are grateful for your encouraging comment. 

Comment 1. I include the review/publication period in the abstract and method section. E.g. in the review, articles published between 2009-2022 were included

Response: Thank you dear reviewer. This systematic review and meta-analysis did not include restriction by year of publication. This was clearly indicated in the methods section under the eligibility of studies sub-section. 

Comment 2. Geographical location of the study. Ethiopia? East Africa? Limit its boundary. 18 studies are from Ethiopia, 3 from migrants and 1 in Kenya. This is good to infer the study findings.

Response: Thank you dear reviewer. Yes, though almost all CL cases (99.9%) in Ethiopia are die to L. aethiopica, case due to this unique species were reported in Mount Elogan area of Kenya, and imported cases with migrants in Israel, Belgium and Germany. So, we did not restrict the studies by geographical location to include studies done on CL due to L. aethiopica in other settings. 

Comment 3. Abbreviation NOS (Non-observational studies?) not clear. Clarify it (page 6)

Response: Thank you dear reviewer. An abbreviation ‘NOS’ refers to ‘Newcastle Ottawa Scale’, and the definition of this abbreviation has been included in the methods section as well as list of abbreviations section of the manuscript. 

Comment 4. Your suggestion and recommendation regarding local therapy for CL due to L. aethiopica.

Response: Thank you dear reviewer. Yes, the findings showed that local therapy had better cure rate, but the use of local therapy was limited to LCL cases. However, combination therapy showed higher cure rate across all clinical phenotypes (LCL, MCL and DCL). 

1.Schulzke S. Assessing and Exploring Heterogeneity. Principles and Practice of Systematic Reviews and Meta-Analysis: Springer; 2021. p. 33-41.

2.Patole S. Systematic reviews, meta-analysis, and evidence-based medicine. Principles and practice of systematic reviews and meta-analysis: Springer; 2021. p. 1-10.

3.Haidich A-B. Meta-analysis in medical research. Hippokratia. 2010;14(Suppl 1):29.

---

## [Decision Letter · Decision Letter 1]

4 Sep 2023

PONE-D-23-03546R1Treatment outcomes of cutaneous leishmaniasis due to Leishmania aethiopica: A systematic review and meta-analysisPLOS ONE

Dear Dr. Abebaw,

Thank you for submitting your manuscript to PLOS ONE. After careful consideration, we feel that it has merit but does not fully meet PLOS ONE’s publication criteria as it currently stands. Therefore, we invite you to submit a revised version of the manuscript that addresses the points raised during the review process.

Please submit your revised manuscript by Oct 15 2023 11:59PM.  If you will need more time than this to complete your revisions, please reply to this message or contact the journal office at plosone@plos.org. Please include the following items when submitting your revised manuscript:A rebuttal letter that responds to each point raised by the academic editor and reviewer(s). You should upload this letter as a separate file labeled 'Response to Reviewers'.A marked-up copy of your manuscript that highlights changes made to the original version. You should upload this as a separate file labeled 'Revised Manuscript with Track Changes'.An unmarked version of your revised paper without tracked changes. You should upload this as a separate file labeled 'Manuscript'.Please include line number while you will  submit  the revised version

We look forward to receiving your revised manuscript.

Kind regards,

Balew Arega Negatie, Msc,MD

Academic Editor

PLOS ONE

Additional Editor Comments:

1.This comment in the previous version must be address“In Figure 5, the is only one study in the subgroup analysis (Seife T/2018). However, there must be at least two studies in each subgroup. The finding of this analysis skewed your conclusion and recommendation. This must be corrected and revise your conclusion accordingly”. You can remove this study from the sub-group MA and do for others. You can narrate about about this study. Dear author, the subgroup meta analysis is just meta analysis, so no possibility to do MA for single study.

2.The manuscript has good progress, but it need extensive language editing. I tried to highlight the most important once.

Reviewers' comments:

Reviewer's Responses to Questions

**Comments to the Author**

1. If the authors have adequately addressed your comments raised in a previous round of review and you feel that this manuscript is now acceptable for publication, you may indicate that here to bypass the “Comments to the Author” section, enter your conflict of interest statement in the “Confidential to Editor” section, and submit your "Accept" recommendation.

Reviewer #2: All comments have been addressed

Reviewer #3: (No Response)

Reviewer #4: (No Response)

2. Is the manuscript technically sound, and do the data support the conclusions?

Reviewer #2: Yes

Reviewer #3: Yes

Reviewer #4: Yes

3. Has the statistical analysis been performed appropriately and rigorously? 

Reviewer #2: Yes

Reviewer #3: Yes

Reviewer #4: Yes

4. Have the authors made all data underlying the findings in their manuscript fully available?

Reviewer #2: Yes

Reviewer #3: Yes

Reviewer #4: Yes

5. Is the manuscript presented in an intelligible fashion and written in standard English?

Reviewer #2: Yes

Reviewer #3: No

Reviewer #4: Yes

6. Review Comments to the Author

Reviewer #2: (No Response)

Reviewer #3: General comments

The authors performed a systematic review and meta-analyzed based on literature searches together evidence on treatment outcome of CL caused by L. aethiopica.

They concluded that combination therapy was the better option to increase the cure rate. However, minor revisions are needed before being considered for publication.

Specific comment

- L3 (Introduction), correct L. tropica writing properly.

- L4 (Introduction), correct L. major writing properly.

- The authors should present a brief introductory paragraph about the life cycle of the parasite describing the primary vectors and the principal reservoir hosts and the abundance of the disease in the area.

- In the discussion section, the authors should describe some of the limitations of the analyzed studies including the lack of clinical trials and analytical studies such as cohort studies and so on.

- A moderate revision of the text is required.

Reviewer #4: See attachment

7. PLOS authors have the option to publish the peer review history of their article (what does this mean?). If published, this will include your full peer review and any attached files.

Reviewer #2: No

Reviewer #3: No

Reviewer #4: Yes, Feleke Tilahun Zewdu

---

## [Author Response · Author response to Decision Letter 1]

25 Sep 2023

Point-by-point response letter

Dears Academic Editor (Dr. Balew Arega Negatie, MD, MSc), Reviewer #2, Reviewer #3 and Reviewer #4, 

Thank you for your constructive comments that we missed in the first round of revision. The authors appreciate your constructive editorial comments that could help to improve the readability and impact of our work. The order of citations in the supporting files S1 Table and S2 Table are updated, and we also uploaded the revised Figure 5.

In this point-by-point response here below, we highlighted the response in light black color below the respective comment (s) by the editor and the reviewers. First, we have presented the comments and the authors’ response to the editor's comments. Then, we presented our responses to the comments by Reviewer #2, Reviewer #3 and Reviewer #4, respectively.

Point-by-point response to editors’ comment

Additional Editor Comments:

Comment 1. This comment in the previous version must be addressed “In Figure 5, there is only one study in the subgroup analysis (Seife T/2018). However, there must be at least two studies in each subgroup. The findings of this analysis skewed your conclusion and recommendation. This must be corrected and revise your conclusion accordingly”. You can remove this study from the sub-group MA and do it for others. You can narrate about this study. Dear author, the subgroup meta-analysis is just meta-analysis, so no possibility of doing an MA for a single study.

Response: Thank you, dear editor. Your comments are highly valuable. We did a meta-analysis after removing the study by Seife et al. and uploaded the revised figure as Fig 5. We also revised the conclusion accordingly. We include all the revisions in the revised manuscript in track changes. The revised subgroup analysis result is included in the revised track change version of the manuscript on page 11, lines 272-280, in the results section. We revised the conclusion in track change in the abstract sections and the conclusions sections accordingly. 

Comment 2. The manuscript has good progress, but it needs extensive language editing. I tried to highlight the most important ones.

Response: Thank you indeed dear editor for editing our work line by line. We forward our heartful thanks for editing our work to make it readable. We have extensively edited the manuscript, and proofread the manuscript. The edits can be seen in track changes from the first page to the end. 

Point-by-point response to Reviewer #2

Comment 1: All comments have been addressed

Response: Dear reviewer # 2, thank you for your review. We forward our sincerest thanks for your time spent reviewing our work and responding to that we have addressed your comments. 

Point-by-point response to Reviewer #3 

General comment

Comment 1: The authors performed a systematic review and meta-analyzed based on literature searches together evidence on treatment outcome of CL caused by L. aethiopica. They concluded that combination therapy was the better option to increase the cure rate. However, minor revisions are needed before being considered for publication.

Response: Thank you dear reviewer for your comment. The manuscript has been revised intensively as per your comments, and the changes made can be seen in the track change in the revised version of the manuscript starting from the first page to the end. 

Specific comment

Comment 1:- L3 (Introduction), correct L. tropica writing properly.

Response: Thank you very much dear reviewer. We have revised as per your comment and the changes included in the track change in line 42 in the introduction section. We have used the italic style because L. tropica is the name of the species. As per the PLOS ONE recommendation, we wrote species name in italic style. 

Comment 2: - L4 (Introduction), correct L. major writing properly.

Response: Response: Thank you very much dear reviewer. We have revised as per your comment and the changes included in the track change on page 2, line 42, in the introduction section. We have used the italic style because L. tropica is the name of the species. As per the PLOS ONE recommendation, we wrote the species name in italic style. 

Comment 4: - The authors should present a brief introductory paragraph about the life cycle of the parasite describing the primary vectors and the principal reservoir hosts and the abundance of the disease in the area.

Response: Thank you very much, dear reviewer. We included a brief statement about the vectors and reservoirs of L. aethiopica in the introduction section. We included the statement: Two hyrax species (Heterohyrax brucei and Procavia capensis), and sandfly (Phlebotomus longipes and Ph. Pedifer) species are the known reservoirs and vectors of L. aethiopica. You can check the changes in the revised version of the manuscript in track changes on page 2, lines 44-46 in the introduction section.

Comment 4: - In the discussion section, the authors should describe some of the limitations of the analyzed studies, including the lack of clinical trials and analytical studies, such as cohort studies and so on.

Response: Thank you indeed, dear reviewer. We have included the limitations of this study and the reviewed studies in the last paragraph of the discussion section of the revised manuscript from on page 15, lines 377-386. 

Comment 5: - A moderate revision of the text is required.

Response: Thank you again for the comment. We have extensively edited the manuscript and proofread it. Dear reviewer, the edits can be seen in track change throughout the entire manuscript from the first page to the end. 

Point-by-point response to Reviewer # 4 

Comment 1: Since you included all CL cases who are treated using both single and combined forms of the treatment. So better to replace it with "treatment outcome".

Response: Thank you reviewer for your comment. We revised the as per the comment, and we include this revision in the revised manuscript in the track change on page 2, line 38, in the abstract section. 

Comment 2: Did you ever check papers using other search engines/strategies like HINARI, CINHAL.....

Response: Thank you so much, dear reviewer. We searched three databases (PubMed, Scopus and ScienceDirect), so we addressed the minimum requirement of three database searches. In addition, CINHAL is not accessible in our country because it requires a subscription. Also, we searched Google Scholar and the reference list of articles to increase the inclusion of potential pocket studies. Furthermore, the two databases, Scopus and PubMed are widely indexed databases in health research, so we believe we did not miss relevant studies. Moreover, articles in HINARI are accessible through PubMed and articles in Embase is accessible in Scopus. 

Comment 3: Sure, you can use this risk of bias assessment tool. But why do not like to use the QOUREM tool for these RCTs?

Response: Thank you, dear reviewer. We chose the revised risk of bias assessment (RoB 2) tool because it is easy to use. The RoB 2 can assess the risk of bias based on five important domains: (1) risk of bias arising from the randomization process, (2) risk of bias due to adherence or assignment to the intended intervention, (3) risk of bias due to missing outcome data, (4) risk of bias in the measurement of outcome, (5) risk of bias in the selection of reported results. The tool is also applicable to all types of RCTs. Additionally, the RoB2 classifies studies into high, medium, and low risk based on the assessors' scores for these five domains. However, we acknowledge the availability of other assessment tools, as you stated clearly. 

Comment 4: Better to change the phrase Clinical cure status and the like

Response: Dear reviewer, thank you for the comment. We revised as per your comment and replaced the wording positive outcome with Favourable outcomes. We use the word favourable instead of clinical cure because in the studies the authors report as “cure, improved, negative/decrease in parasite density, good response, decrease in the size of the lesion, and resolution of the lesion without scar”. Hence, we use favourable outcomes instead of positive outcomes. The change can be seen in the revised version of the manuscript in track change, starting from page 7 line 179, in the results section. 

Comment 5: What do you feel if someone says negative outcome is: something comes secondary to a certain agent, adverse event and the like? So better to replace it with another better descriptive word, please. Like "Nil, poor, ...... responsive ". 

Response: Thank you indeed, dear reviewer. We have revised as per your comment and replaced the word negative outcomes with Unfavourable outcomes. The changes can be seen in track change in the revised version, starting from page 9, line 226, in the results section. The word unfavourable was deemed more descriptive for the authors because the studies we reviewed report a range of outcomes such as recurrence, resistance, treatment extension, partial response, relapse, dropout, and unresponsiveness, progression to a more severe stage and toxicity. So, we chose the word unfavourable which we believe best described these outcomes. 

Comment 6: Include the limitation of the study.

Response: Thank you, dear reviewer. We have included the limitation of this study and the studies we included in this review in the last paragraph of the discussion section, page 15, from lines 377-386. The response to these comments is included in the track change in the revised version of the manuscript.

---

## [Decision Letter · Decision Letter 2]

10 Oct 2023

PONE-D-23-03546R2Treatment outcomes of cutaneous leishmaniasis due to Leishmania aethiopica: A systematic review and meta-analysisPLOS ONE

Dear Dr. Alemu,

Thank you for submitting your manuscript to PLOS ONE. After careful consideration, we feel that it has merit but does not fully meet PLOS ONE’s publication criteria as it currently stands. Therefore, we invite you to submit a revised version of the manuscript that addresses the points raised during the review process.

Kind regards,

Balew Arega Negatie, Msc,MD

Academic Editor

PLOS ONE

Journal Requirements:

Reviewers' comments:

Reviewer's Responses to Questions

**Comments to the Author**

1. If the authors have adequately addressed your comments raised in a previous round of review and you feel that this manuscript is now acceptable for publication, you may indicate that here to bypass the “Comments to the Author” section, enter your conflict of interest statement in the “Confidential to Editor” section, and submit your "Accept" recommendation.

Reviewer #3: All comments have been addressed

2. Is the manuscript technically sound, and do the data support the conclusions?

Reviewer #3: Yes

3. Has the statistical analysis been performed appropriately and rigorously? 

Reviewer #3: Yes

4. Have the authors made all data underlying the findings in their manuscript fully available?

Reviewer #3: Yes

5. Is the manuscript presented in an intelligible fashion and written in standard English?

Reviewer #3: Yes

6. Review Comments to the Author

Reviewer #3: L45, the name of the species should be written in small font (e.g., Ph. pedifer).

L113, Leishmania aethiopica should be italicized.

L355, Leishmania should be written correctly ( the “L” should be capitalized).

7. PLOS authors have the option to publish the peer review history of their article (what does this mean?). If published, this will include your full peer review and any attached files.

Reviewer #3: **Yes: **Iraj Sharifi

---

## [Author Response · Author response to Decision Letter 2]

13 Oct 2023

Point-by-point response letter

Dears Academic Editor (Dr. Balew Arega Negatie, MD, MSc) and Reviewer #3 (Professor Iraj Sharifi, PhD, Managing Director at Kerman University of Medical Sciences, Iran)

Thank you for your constructive comments that we missed in the second round of the revision. The authors forward heartfelt gratitude for your scholarly insights and comments to increase the readability of our manuscript. 

In this point by point response letter, first, we respond to the query on journal requirements about the reference list in the manuscript. Followed by point-by-point response for the comments by Reviewer #3. In this point-by-point response here below, we highlighted the author response in light black color below the respective comment (s) by the reviewer.

Point by point response to the Journal Requirements query

Comment: Please review your reference list to ensure that it is complete and correct. If you have cited papers that have been retracted, please include the rationale for doing so in the manuscript text, or remove these references and replace them with relevant current references. Any changes to the reference list should be mentioned in the rebuttal letter that accompanies your revised manuscript. If you need to cite a retracted article, indicate the article’s retracted status in the References list and also include a citation and full reference for the retraction notice.

Author response: Thank you dear editor. We reviewed the reference list are complete and correct, and ensured that no reference (s) cited were retracted. In addition, all the references are based on the Vancouver style as per the PLOS ONE requirement. 

Point by point response to reviewer comments

Comments to the Author

1. If the authors have adequately addressed your comments raised in a previous round of review and you feel that this manuscript is now acceptable for publication, you may indicate that here to bypass the “Comments to the Author” section, enter your conflict of interest statement in the “Confidential to Editor” section, and submit your "Accept" recommendation.

Reviewer #3: All comments have been addressed

Author response: Thank you, dear reviewer, for replying that we have addressed your comments. 

2. Is the manuscript technically sound, and do the data support the conclusions?

Reviewer #3: Yes

Author response: Thank you, dear reviewer, for your encouraging response and valuing our efforts. 

3. Has the statistical analysis been performed appropriately and rigorously?

Reviewer #3: Yes

Author response: Dear reviewer, thank you again for your encouraging response and valuing our efforts of using rigorous methodology.

4. Have the authors made all data underlying the findings in their manuscript fully available?

Reviewer #3: Yes

Author response: Dear reviewer, thank you indeed. 

5. Is the manuscript presented in an intelligible fashion and written in standard English?

Reviewer #3: Yes

Author response: Dear reviewer, thank you again for your encouraging response and valuing our efforts of reviewing to meet the standard English language requirements

6. Review Comments to the Author

Review comment 1: L45, the name of the species should be written in small font (e.g., Ph. pedifer).

Author response: Thank you, dear reviewer, for reading our work line by line. We revised the species name as per your comment and the change can be seen on line 45, page 2, in the tracked change version of the manuscript.

Review comment 2: L113, Leishmania aethiopica should be italicized.

Author response: Thank you, dear reviewer, for reading our work line by line. We revised the species name as per your comment and the change can be seen on line 113, page 5, in the tracked change version of the manuscript.

Review comment 3: L355, Leishmania should be written correctly (the “L” should be capitalized).

Author response: Thank you, dear reviewer, for reading our work line by line. We revised the species name as per your comment and the change can be seen on line 355, page 14, in the tracked change version of the manuscript.

---

## [Editor Report · Decision Letter 3]

16 Oct 2023

Treatment outcomes of cutaneous leishmaniasis due to Leishmania aethiopica: A systematic review and meta-analysis

PONE-D-23-03546R3

Dear Dr. Abebaw

We’re pleased to inform you that your manuscript has been judged scientifically suitable for publication and will be formally accepted for publication once it meets all outstanding technical requirements.

Kind regards,

Balew Arega Negatie, Msc,MD

Academic Editor

PLOS ONE
---

## [Editor Report · Acceptance letter]

24 Oct 2023

PONE-D-23-03546R3 

Treatment outcomes of cutaneous leishmaniasis due to Leishmania aethiopica: A systematic review and meta-analysis 

Dear Dr. Alemu:

I'm pleased to inform you that your manuscript has been deemed suitable for publication in PLOS ONE. Congratulations! Your manuscript is now with our production department. 

Kind regards, 

on behalf of

Dr. Balew Arega Negatie 

Academic Editor

PLOS ONE